# Physicochemical Properties and Elimination of the Activity of Anti-Nutritional Serine Protease Inhibitors from Mulberry Leaves

**DOI:** 10.3390/molecules27061820

**Published:** 2022-03-11

**Authors:** Zhuxing Luo, Jinhong Yang, Jie Zhang, Gang Meng, Qingjun Lu, Xi Yang, Ping Zhao, Youshan Li

**Affiliations:** 1College of Biological Science and Engineering, Shaanxi University of Technology, Hanzhong 723001, China; luozhx@snut.edu.cn (Z.L.); zhangjie@snut.edu.cn (J.Z.); luqj@snut.edu.cn (Q.L.); yangxi@snut.edu.cn (X.Y.); 2Shaanxi Key Laboratory of Sericulture, Ankang University, Ankang 725099, China; yangjinhong@aku.edu.cn (J.Y.); nsymg@aku.edu.cn (G.M.); 3State Key Laboratory of Silkworm Genome Biology, Southwest University, Chongqing 400715, China; zhaop@swu.edu.cn

**Keywords:** *Morus*, anti-nutritional factor, serine protease inhibitor, physicochemical properties, inactivation methods

## Abstract

Mulberry leaf is an excellent protein resource that can be used as feed additive for livestock and poultry. Nevertheless, the use of mulberry leaves in animal diets is limited by its protease inhibitors, tannic acid and other anti-nutritional factors. This study systematically analyzed the type and activity of serine protease inhibitors (SPIs) from the leaves of 34 mulberry varieties, aiming to reveal the physicochemical properties and inactivation mechanism of SPIs. The types and activities of trypsin inhibitors (TIs) and chymotrypsin inhibitors (CIs) exhibited polymorphisms among different mulberry varieties. The highest number of types of inhibitors was detected in Jinshi, with six TIs (TI-1~TI-6) and six CIs (CI-1~CI-6). TIs and CIs exhibited strong thermal and acid–base stability. High-temperature and high-pressure treatment could reduce the activities of TIs and CIs to a certain extent. β-mercaptoethanol treatment could completely abolish TIs and CIs, suggesting that the disulfide bridges were critical for their inhibitory activities. The Maillard reaction could effectively eliminate the inhibitory activities of TI-1~TI-4 and CI-1~CI-4. This study reveals the physicochemical properties and inactivation mechanisms of the anti-nutritional SPIs from mulberry leaves, which is helpful to exploit mulberry-leaf food with low-activity SPIs, promote the development and utilization of mulberry-leaf resources in animal feed and provide reference for mulberry breeding with different functions.

## 1. Introduction

Mulberry, belonging to the Moraceae family, is a deciduous tree or shrub. China has the most abundant germplasm resources of mulberry in the world, with 15 species and 4 varieties. Mulberry can also be divided into two groups, with *Morus multicaulis*, *Morus alba*, *Morus atropurpurea* and *Morus mizuho* being cultivated species and *Morus cathayana*, *Morus nigra*, *Morus leavigata*, *Morus bombycis*, etc., being wild species [1]. The mulberry leaf has already been listed in the first batch of “medicine and food homology” by the Ministry of Health of China in 1993. Mulberry leaves contain alkaloids, flavonoids, polysaccharides, polyphenols and other chemical components which endow mulberry leaves with a wide range of pharmacological activities, such as anti-cancer, anti-virus and anti-obesity activities; they also lower blood sugar and prevent arteriosclerosis [2,3,4,5,6]. Traditionally, mulberry leaves have mainly been used to feed silkworms. Recently, mulberry leaves have become high-quality animal feed due to their high yield, low cost, good palatability and abundant protein content. As the feed of pigs, sheep, chicken broilers and other livestock, mulberry leaves can improve the conversion efficiency of feed nutrition in animals and enhance the quality of livestock and poultry products [7,8,9]. This not only alleviates the shortage of high-quality forage resources to a certain extent but also avoids the waste of mulberry-leaf resources and increases the comprehensive economic benefits of the sericulture industry. However, mulberry leaves may contain protease inhibitors, tannic acid, lectins and other anti-nutritional factors [10,11]. Large additions would seriously interfere with the metabolism and absorption of feed nutrients in animals, consequently affecting animal health and the yield and quality of livestock and poultry products, which would greatly limit the development and large-scale application of mulberry leaves resources in animal feed [8,12,13].

Serine protease inhibitors (SPIs) are the most numerous and most thoroughly studied protease inhibitors, including trypsin inhibitors (TIs), chymotrypsin inhibitors (CIs), elastase inhibitors (EIs), subtilisin inhibitors (SIs), etc. Plant SPIs are essential for protection against pathogen infection and insect-feeding hazards [14,15]. Based on the known protease inhibitor (PI) sequence and its conserved domains, 79 PIs were identified in the genome of *Morus notabilis*, including 35 SPIs, which were classified as Kunitz, Serpin and PI-I family [16,17,18]. It was found that eight Kunitz and one Serpin genes were mainly expressed in mulberry leaves. It is speculated that these SPIs may play an important role in the defense of mulberry trees against insect-feeding hazards [18]. Experiments simulating insects biting leaves found that the expression of genes in the PI-I family could respond to leaf trauma signals, suggesting the presence of pest-defense SPI genes in mulberry leaves [18]. Wang D. et al. detected multiple activity-stained bands of CIs from white milk extracted from mulberry petioles by in-gel activity staining, but CI activity was not detected in mulberry leaves [19]. At present, there are no reports on anti-nutritional SPIs from mulberry leaves; the activities, physicochemical properties and inactivation methods of these SPIs are still unclear.

In this study, we analyzed the types, activity polymorphisms and physicochemical properties of TIs, CIs, EIs and SIs from the leaves of 34 mulberry varieties and explored the inactivation mechanisms of TIs and CIs from mulberry leaves. Such knowledge would enhance the nutrient utilization of mulberry leaves, provide a theoretical basis and new approaches for the development of mulberry-leaf products with low-activity SPIs and promote the exploitation and utilization of mulberry-leaf resources in animal feed.

## 2. Results

### 2.1. TI and CI Activities Were Present in Mulberry Leaves

Animal intestine is the major site for protein digestion and absorption. Serine proteases such as trypsin, chymotrypsin and elastase play a key role in the deep hydrolysis of food proteins. Therefore, we speculate that some SPIs from mulberry leaves may act as important anti-nutritional factors affecting protein digestion and absorption in animals. In order to explore whether anti-nutritional TI and CI inhibitory activities were present in mulberry leaves, cultivated mulberry Guang1 was randomly selected as the research object. TI and CI activities in mulberry leaves were detected by alkaline Native PAGE and in-gel activity staining of SPIs. The results show that four TI activity-stained bands (Figure 1B) and four CI activity-stained bands (Figure 1C) were detected in Guang1, implying that at least four TIs and four CIs existed in the analyzed mulberry leaves.

### 2.2. Types and Activity Distribution of TIs from Mulberry Leaves

In order to systematically explore the types and activity polymorphisms of TIs from the leaves of different mulberry varieties, the mulberry-leaf proteins of 34 varieties were separated by alkaline and acidic Native PAGE and TI activity was analyzed by in-gel activity staining. The mulberry varieties used in the experiments are shown in Table 1. In total, six TI activity-stained bands were detected by alkaline Native PAGE and were named TI-1~TI-6 from top to bottom (Figure 2A). Moreover, the types and activity of TIs exhibited polymorphisms among different mulberry varieties. No TI activity was detected in acidic Native PAGE (data not shown). The highest number of types of TIs was detected in the leaves of Dongguangdabai, Zhenzhubai and Jinshi, with a total of six TIs, including TI-1~TI-6. The activities of TI-2~TI-6 were detected in Luyou3, Guang1 and Shaosang. Four TIs, TI-3~TI-6, were mainly detected in Sichuantiansang, Luyou7, Hongguo2, Shi2, Guoshensang, Xiaoguansang, Shaansang403, Xiaohongpi, Xinjiangbaisang and Yixianheisang. The activities of TI-5 and TI-6 were mainly detected in Guangdongzajiao and Guiyou62. Weak TI activity was detected in Nong12, Dashi, Beiqu1, Shalun, Huaiyin4, Tuosang, Ukrainian1, Hulusang, Taiwanchangguosang and Kenmochi. No TI activity was detected in the leaves of Nong14, Heiyumodou, Yu711, Lunjiao408, Guiyou12 and Qingyeshufan. Among the tested *M. alba* varieties, TI activity was detected in the majority of mulberry varieties. Compared with other varieties, the inhibitory activity of TIs was the strongest in the leaves of Luyou3 and Shaansang403, followed by Guang1 and Jinshi. The above results show that TI activity was present in mulberry leaves and the isoelectric points of TIs were less than 8.3.

### 2.3. Types and Activity Distribution of CIs in Mulberry Leaves

In order to further explore the types and activity polymorphisms of CIs from the leaves of different mulberry varieties, the activity of CIs was detected in mulberry leaves by in-gel activity staining. In total, six CIs were detected in mulberry leaves by alkaline Native PAGE and were named CI-1~CI-6 from top to bottom (Figure 2B). The types and activity of CIs also exhibited polymorphisms among different mulberry varieties. In the acid electrophoresis, no CI activity was detected in mulberry leaves (data not shown). The most abundant variety of CIs, including CI-1~CI-6, was detected in the leaves of Dongguangdabai, Zhenzhubai and Jinshi. Five CIs, CI-2~CI-6, were detected in Luyou3, Guang1 and Shaosang. Four CIs, CI-3~CI-6, were mainly expressed in the leaves of 10 mulberry varieties, including Sichuantiansang, Luyou7, Hongguo2, Shi2, Guoshensang, Xiaoguansang, Shaansang403, Xiaohongpi, Xinjiangbaisang and Yixianheisang. The activity of CI-6 was mainly detected in Guangdongzajiao and Guiyou62. Only weak CI bands were detected in Nong12, Dashi, Beiqu1, Shalun, Huaiyin4, Tuosang, Ukrainian1, Hulusang, Taiwanchangguosang and Kenmochi. The inhibitory activity against chymotrypsin was not detected in the mulberry leaves of other varieties. CI activity was detected in the leaves of most of the tested *M. alba* varieties. Compared with other varieties, the inhibitory activity of CIs was the highest in Luyou3 and Shaansang403, followed by Guang1 and Jinshi. It should be noted that the electrophoretic mobilities of CI-1~CI-6 and TI-1~TI-6 were nearly the same, in accordance with the position of the inhibitor activity-stained bands in alkaline Native PAGE, and the variation rules of activity intensity among different varieties was also nearly the same. It was suggested that CI-1~CI-6 and TI-1~TI-6 may correspond to the same proteins, respectively.

### 2.4. No Elastase Inhibitor nor Subtilisin Inhibitor Activity Was Detected in Mulberry Leaves

Besides TI and CI, are other SPIs such as elastase inhibitors and subtilisin inhibitors present in mulberry leaves? To answer this question, we tested the leaves of 34 mulberry varieties using Native PAGE and in-gel activity staining for the presence of elastase inhibitors and subtilisin inhibitors. The results show that no elastase inhibitors (Appendix A) nor subtilisin inhibitors (Appendix A) were detected in the leaves of the tested mulberry varieties.

### 2.5. TIs and CIs from Mulberry Leaves Had Strong Acid–Base and High Thermal Stability

In view of the strongest TI and CI activities in the leaves of mulberry varieties Guang 1 and Jinshi, the two varieties were selected for the subsequent study of physicochemical properties and inactivation methods. To determine the acid–base stability of TIs and CIs from mulberry leaves, we analyzed the protease-inhibitor activity in mulberry-leaf proteins treated under different pH conditions by alkaline Native PAGE and in-gel activity staining. The results show that the TIs and CIs (Figure 3A,B) from the leaves of Guang1 and Jinshi exhibited strong acid–base stability over a wide pH range (pH 3~11). In the two varieties, both TIs and CIs showed the strongest inhibitory activities at pH 9.0. In general, the inhibitory activities of TIs and CIs were stronger in the alkaline environment than in the acidic environment. In the acidic environment (pH 3~5), the activities of TI-6 and CI-6 lowered sharply with the decrease in pH, indicating that TI-6 and CI-6 were less stable under acidic conditions. The above findings indicate that TIs and CIs from mulberry leaves exhibited strong acid–base stability and the activities of TIs and CIs could not be effectively reduced by altering the pH of the external environment.

It is generally considered that heat treatment is one of the most effective ways to eliminate the anti-nutritional factors found in plants. In the processing of mulberry-leaf powder, mulberry leaves with different maturity are usually blanched in water at different temperatures for 5 min—young leaves at 60 °C, mature leaves at 85 °C and old leaves (also known as frost leaves) at 95 °C. Combined with the processing technology of mulberry-leaf powder, the leaf proteins of Guang1 and Jinshi were treated at 95 °C for different lengths of time to analyze the thermal stability of TIs and CIs from mulberry leaves and explore the inactivation methods of anti-nutritional SPIs. The results show that the activities of TIs and CIs (Figure 3C,D) from mulberry leaves were not significantly reduced after heating at 95 °C for different lengths of time, indicating that both TIs and CIs from mulberry leaves had high thermal stability.

### 2.6. Combined Treatment of High Temperature and High Pressure Could Greatly Weaken TI and CI Activities in Mulberry Leaves

In order to inactivate the TIs and CIs in mulberry leaves, the temperature was further raised to 100 °C, or the mulberry-leaf proteins were treated by autoclaving at 121 °C for 20 min (Figure 4A,B). The results showed that, compared with the control group, the activities of TIs and CIs showed little change after treatment at 100 °C for 20 min. The activities of TIs and CIs could be greatly impaired at 121 °C and 0.21 MPa for 20 min, suggesting that the combination of high temperature and high pressure could be used as an effective way to inactivate anti-nutritional SPIs from mulberry leaves.

### 2.7. TI and CI Activities in Mulberry Leaves Could Be Eliminated by Treating with β-Mercaptoethanol

Generally, cysteine-rich protease inhibitors have high thermal and acid–base stability [20]. Does the high thermal and acid–base stability of TIs and CIs from mulberry leaves also depend on disulfide bridges? Mulberry-leaf proteins were heated or not heated in the presence or absence of the reducing agent β-mercaptoethanol (Figure 5A,B). The results show that heat treatment had little effect on TI and CI activities in the absence of β-mercaptoethanol. All activities of TIs and CIs were greatly weakened in the presence of β-mercaptoethanol without heat treatment. When heat treatment was performed in the presence of β-mercaptoethanol, all activities of TIs and CIs were completely lost. The above results indicate that disulfide bridges existed in the structures of both TI-1~TI-6 and CI-1~CI-6 in the analyzed mulberry leaves and that disulfide bridges were critical for the maintenance of their activities. The destruction of disulfide bridges was an effective way to eliminate anti-nutritional SPIs from mulberry leaves.

### 2.8. Glucose-Mediated Maillard Reaction Impaired Part of TI and CI Activities

The P1 residue in the TI reaction center is usually the basic amino acid Arg or Lys. Modifying the free amino group of the P1 residue by reducing the sugar-mediated Maillard reaction may be able to reduce TI activity in mulberry leaves [21]. So, mulberry-leaf proteins were heated or not heated with reductive monosaccharides to eliminate the activity of TIs. The activities of TIs and CIs from Guang1 and Jinshi were nearly unaffected in the presence of glucose but without heat treatment (Figure 6A,B). Heat treatment in the presence of glucose could accelerate the Maillard reaction, which, in turn, caused complete loss of the inhibitory activities of TI-1 and TI-2 and a large reduction in the activities of TI-3 and TI-4, and decreased the activities of TI-5 and TI-6 to a certain extent. The activities of TI-1~TI-6 also decreased slightly after heating for 60 min without exogenous glucose, which may be related to the Maillard reaction mediated by endogenous reducing sugars, aldehydes and ketones in mulberry leaves. It should be noted that the Maillard reaction also simultaneously reduced the activities of CI-1~CI-6 and the decreasing degree of activity was consistent with the activity change trend of TI-1~TI-6.

## 3. Discussion

Herein, we demonstrated, for the first time, the existence of multiple TI and CI activity-stained bands in mulberry leaves, with TIs and CIs showing strong acid and alkali resistance and high thermal stability. Moreover, high-temperature and high-pressure treatment and the glucose-mediated Maillard reaction could inactivate the protease inhibitors to varying degrees. Treatment of mulberry-leaf proteins with a reducing agent caused complete loss of TIs and CIs from mulberry leaves. Reducing-agent treatment could completely abolish the activities of TIs and CIs from mulberry leaves.

Plant SPIs can help resist insect feeding and pathogen infection, but they are not conducive to the digestion and absorption of nutrients in animals. Eight Kunitz and one Serpin genes were mainly expressed in leaves, but no TI or CI activity was detected in mulberry leaves [16,18,19]. Is SPI activity present in mulberry leaves? What are their physicochemical properties and inactivation mechanisms? In this study, in total, 34 mulberry varieties were selected to systematically analyze the activities of SPIs from mulberry leaves. We identified that there were at least six TI and six CI activity-stained bands in mulberry leaves. Previous studies did not detect TI or CI activity in the leaves of *M. notabilis*, which may be caused by differences in mulberry varieties, protein extraction techniques and other factors. The activity-stained bands of TIs and CIs were detected only in alkaline gel, indicating that the isoelectric points of TIs and CIs from mulberry leaves were less than 8.3. Hao H. and Wang Y. found that the distribution of the theoretical isoelectric points of 35 SPIs from *M. notabilis* tended to be acidic and only 3 SPIs had isoelectric points greater than 8.3, consistently with the results of this study [16,18].

Trypsin and chymotrypsin are the main intestinal proteases. Their inhibitors compete with nutrients for protease action sites, thus hindering the binding of intestinal nutrients to proteases, so as to inhibit digestion. With the improvement of people’s living standards and the change in the dietary structure, simple obesity caused by excessive intake of high-fat, high-protein or high-sugar foods is one of the chronic diseases faced by adults. The study of high-fat-diet-fed mice or obese mice fed with anti-nutritional TIs extracted from soybean whey found that the weight of high-fat-diet-fed mice did not increase significantly, while the weight of obese mice decreased significantly and the indexes of triglycerides, high-density lipoproteins and free fatty acids were significantly improved [22]. This indicates that TIs could prevent and treat nutritional obesity. Mulberry leaves, as medicinal and edible homologous plants, also have the effect of weight loss. A study of high-fat-diet-induced obese mice fed with mulberry leaves found that mulberry leaves could reduce weight gain, fat accumulation and fasting blood sugar of mice and improve insulin sensitivity by enhancing the activity of brown adipose tissue, suggesting that mulberry leaves have therapeutic potential for obesity [5]. The ethanol extract of mulberry leaves was also reported to have antiadipogenic effects on differentiated adipocytes, suggesting that mulberry-leaf ethanol extract is also effective in preventing obesity [23]. So, do water-soluble PIs also play a role in the prevention and treatment of obesity? Our study found that TI and CI activities exhibited polymorphisms among the leaves of different mulberry varieties. There were differences in the activity levels of protease inhibitors across different mulberry varieties in the same species, which might be due to the large genetic differences among different mulberry varieties. A study showed that mulberry germplasm resources are extremely rich and complex, which also results in different traits among different mulberry plants, such as disease resistance, insect resistance, stress resistance, etc. [24]. TI and CI activities in the leaves of Luyou3 and Shaansang403 were the strongest, suggesting that the mulberry leaves of these two varieties may have more advantages in weight loss, which could provide a new perspective for the development of mulberry-leaf slimming tea and mulberry breeding. No TI or CI inhibitory bands were detected in the leaves of Nong14, Heiyumodou, Yu711, Lunjiao408, Guiyou12 and Qingyeshufan, indicating that the expression levels of anti-nutritional TIs and CIs were very low, implying that these mulberry varieties may have great potential in the development of feed for livestock and poultry.

TI-1~TI-6 and CI-1~CI-6 from mulberry leaves exhibited strong acid–base and thermal stability. This finding is consistent with the results of previous studies on PIs isolated from the seeds of *Cajanus cajan*, *Clitoria fairchildiana* and *Rhynchosia sublobata* [25,26,27]. These inhibitors maintained stable inhibitory activity against trypsin and chymotrypsin after heat treatment at 100 °C for 30 min and also showed strong acid and alkali resistance.

Heat treatment is a traditional way to destroy anti-nutritional factors in many foods [28]. The combination of high temperature and humidity in food cooking is a proven way to effectively weaken the activities of TIs and CIs in food. Heating inactivation of PIs is not only related to the physicochemical properties of the inhibitors, but it is also related to the heating method, heating temperature, heating time, etc. To eliminate PIs from mulberry leaves, we further increased the temperature and improved the treatment conditions. The results show that the combined treatment of high temperature and high pressure could weaken the activities of TIs and CIs from mulberry leaves to a certain extent. Similar to this study, TI and CI activities in the seeds of *Mucuna pruriens* and *Sorghum bicolo* were significantly decreased after high-pressure cooking [21,28].

High temperature and prolonged heating often lead to the reduction in nutrients and sensory quality of food, so a relevant research direction is investigating how to eliminate PI activity to the maximum extent without destroying food nutrients. In general, cysteine-rich protease inhibitors have strong thermal and acid–base stability. Our previous studies showed that the cysteine-rich protease inhibitors BmSPI38 and BmSPI39 in *Bombyx mori* were very stable over a wide range of temperature and pH values [20,29]. This property was also fully confirmed in the PIs purified from the seeds of *Cajanus cajan*, *Rhynchosia sublobata* and *Vigna mungo* [25,27,30]. In view of this, we sought to treat mulberry-leaf proteins with a reducing agent to eliminate TI and CI activities in mulberry leaves. TI and CI activities in mulberry leaves were completely lost after treatment with β-mercaptoethanol, indicating that the disulfide bridge played a critical role in maintaining the stability of TI and CI activities in mulberry leaves and that destroying the disulfide bridge was an effective means to eliminate SPI activities in mulberry leaves.

TI activity in mulberry leaves was decreased to different extents after treatment with the glucose-mediated Maillard reaction. Surprisingly, CI activity in mulberry leaves was also weakened to a similar degree. Recently, glucose-mediated Maillard reactions were also reported to have weakened TI and CI activities in soy milk [31]. However, He H. et al. found that TI and CI activities were increased after treating soybean milk with glucose [32], speculating that the newly generated products in the reaction system might interfere with the detection results of TI and CI activities. In this study, the interference of small molecular compounds produced by the Maillard reaction on the experimental results could be eliminated by in-gel activity staining of SPIs.

According to the position of the active bands of protease inhibitors in alkaline Native PAGE gel, it could be seen that the electrophoretic mobilities of TI-1~TI-6 and CI-1~CI-6 were nearly the same and the variation rules of activity intensity among different varieties were also nearly the same. Moreover, the activity trends of TI-1~TI-6 and CI-1~CI-6 from mulberry leaves were almost the same after treatment with acidic–basic pH, high temperature and high pressure, the reducing agent or the glucose-mediated Maillard reaction. These results suggest that TI-1~TI-6 and CI-1~CI-6 may correspond to the same proteins in mulberry leaves, respectively. Similar to our study, the protease inhibitors that could inhibit both trypsin and chymotrypsin were isolated and purified from the leaves of *Moringa oleifera* and *Coccinia grandis* [33,34]. It should be pointed out that the structure and function of anti-nutritional SPIs from mulberry leaves need further study. We will conduct mass spectrometry identification of anti-nutritional SPIs from mulberry leaves and study their anti-nutritional effects in animals.

## 4. Materials and Methods

### 4.1. Mulberry-Leaf Collection and Main Reagents

In total, 34 mulberry varieties, including 30 cultivated species and 4 wild species, were cultivated at Hengkou Sericulture Research Base, Ankang University, China. Mulberry leaves were picked on 19 August 2020. The maturity of the mulberry leaves was approximately the same. Trypsin, chymotrypsin, subtilisin and N-acetyl-D,L-phenylalanine-β-Naphthalene ester (APNE) were purchased from Sigma. Elastase was obtained from Sangon Biotech.

### 4.2. Extraction and Quantification of Mulberry-Leaf Protein

Fresh mulberry leaves were washed, drained and then ground into powder in liquid nitrogen. Mulberry-leaf powder was mixed with 0.1 mol/L Tris-HCl buffer and incubated for 1 h at 4 °C. After centrifugation at 4 °C (12,000× *g* for 30 min), the supernatant was sonicated for 15 min and centrifuged again. The supernatant was concentrated using a 3 kDa MWCO ultrafiltration tube. The Bradford method was used for protein quantification [35].

### 4.3. Native PAGE

The 60, 80, 120 and 140 µg mulberry-leaf protein samples were mixed with 4 × Native PAGE buffer. Then, they were separated via 10% Native PAGE under alkaline (Tris-Glycine buffer, pH 8.3) or acidic (Alanine-HAc buffer, pH 4.0) conditions.

### 4.4. In-Gel Activity Staining of SPIs

The activity staining of SPIs from mulberry leaves was performed as previously described, with a slight modification [20]. After separating with 10% Native PAGE, the gels were incubated in 0.07 mg/mL trypsin, 0.07 mg/mL chymotrypsin, 0.07 mg/mL subtilisin or 5 mg/mL elastase solution (0.1 mol/L Tris-HCl, 20 mM CaCl_2_, pH 8.0) in the dark for 30 min at 37 °C, with shaking at 45 rpm. Then, the protease solution was recovered, the gels were washed with ddH_2_O and were allowed to stand at 37 °C for 30 min. The gels were stained for 15 min with a mixture of substrates (20 mg of APNE dissolved in 10 mL of N, N’-dimethylformamide) and a staining solution (100 mg of Fast Blue B Salt dissolved in 100 mL of 0.1 mol/L Tris-HCl, 20 mM CaCl_2_, pH 8.0) at a ratio of 1:10 at 37 °C in the dark. The gels were stained fuchsia due to the diazotization-coupling reaction of β-naphthol that was produced by protease hydrolyzed substrates (APNE) on the gel. Protease inhibitors in the gel could inhibit the activity of the corresponding protease; therefore, white bands appeared at the positions where the protease inhibitors were present (Figure 1A).

### 4.5. Acid–Base and Thermal Stability of TIs and CIs from Mulberry Leaves

First, 80 µg mulberry-leaf protein samples were mixed with equal volumes of Britton–Robinson buffer at different pH values and incubated at room temperature for 24 h. The same protein samples were treated at 95 °C for different lengths of time. Then, the samples were separated using alkaline Native PAGE. Finally, the activities of TIs and CIs were analyzed using in-gel activity staining of SPIs.

### 4.6. Elimination of Activities of TIs and CIs from Mulberry Leaves

Combined treatment of high temperature and high pressure: Eighty-microgram mulberry-leaf protein samples were treated at 100 °C or 121 °C (0.21 MPa) for 20 min. Treatment with β-mercaptoethanol: The same protein samples were mixed with reducing or nonreducing Native PAGE buffer (with or without 8% β-mercaptoethanol) and then boiled or not boiled for 10 min. Glucose-mediated Maillard reaction: The same protein samples were mixed with equal volumes of 15 mg/mL glucose solution or ultrapure water and then heated for 60 min at 100 °C or left to stand at room temperature. Finally, the activities of TIs and CIs were analyzed by alkaline Native PAGE and in-gel activity staining of SPIs.

## 5. Conclusions

In summary, the study confirms, for the first time, that at least six TI and six CI activity-stained bands exist in mulberry leaves and reveals their physicochemical properties and inactivation mechanisms. This not only provides a new perspective and idea for the exploitation and utilization of mulberry leaves in animal feed and healthy food, but it also has a certain guiding significance for mulberry breeding.

## Figures and Tables

**Figure 1 molecules-27-01820-f001:**
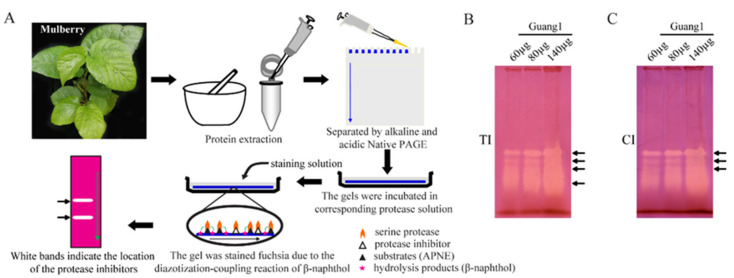
The principle of in-gel activity staining and detection of TI and CI activities in mulberry leaves. (**A**) Schematic diagram of in-gel activity staining of SPIs. Activity staining of TIs (**B**) and CIs (**C**) from leaves of mulberry variety Guang1. “TI” and “CI” represent trypsin inhibitor and chymotrypsin inhibitor, respectively. The arrows show activity-stained bands of protease inhibitors.

**Figure 2 molecules-27-01820-f002:**
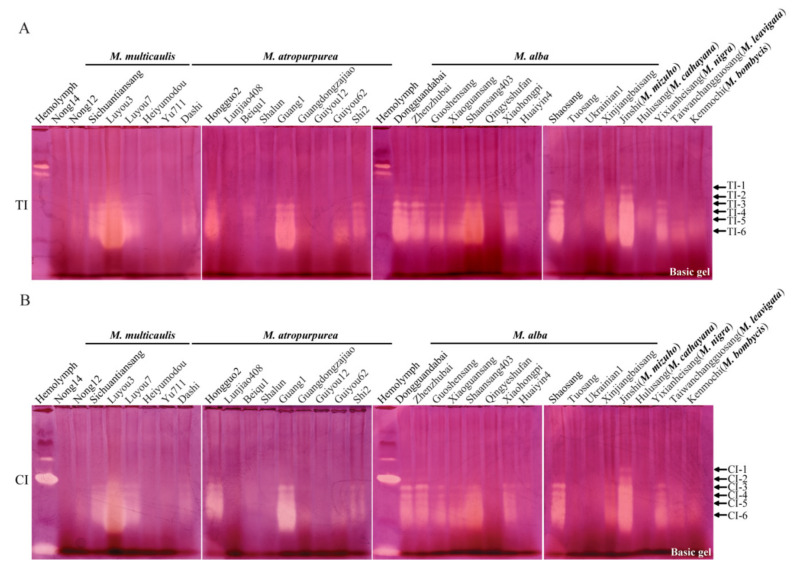
Activity staining of TIs (**A**) and CIs (**B**) from leaves of different mulberry varieties. “Basic gel” indicates Native PAGE under alkaline conditions. “TI” and “CI” represent trypsin inhibitor and chymotrypsin inhibitor, respectively. *B. mori* hemolymph from day-5 fifth-instar larvae was used as positive control. From left to right, varieties from “Nong14” to “Yu711” belong to *M. multicaulis*; varieties from “Dashi” to “Shi2” belong to *M. atropurpurea*; varieties from “Dongguangdabai” to “Xinjiangbaisang” belong to *M. alba*. The last five mulberry varieties belong to *M. mizuho*, *M. cathayana*, *M. nigra*, *M. leavigata* and *M. bombycis*, respectively. The arrows show activity-stained bands of trypsin inhibitors and chymotrypsin inhibitors, which are named “TI-1~TI-6” and “CI-1~CI-6” from top to bottom.

**Figure 3 molecules-27-01820-f003:**
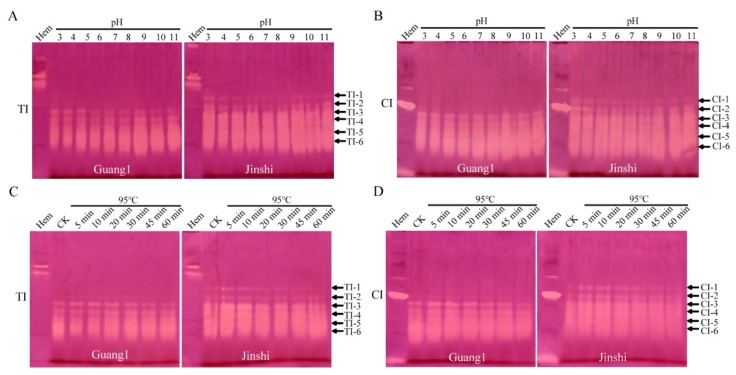
TIs and CIs from the leaves of Guang1 and Jinshi exhibited strong acid–base and thermal stability. Acid–base stability of TIs (**A**) and CIs (**B**) from leaves of mulberry varieties Guang1 and Jinshi. Thermal stability of TIs (**C**) and CIs (**D**) from leaves of mulberry varieties Guang1 and Jinshi. “TI” and “CI” represent trypsin inhibitor and chymotrypsin inhibitor, respectively. *B. mori* hemolymph from day-5 fifth-instar larvae represented by “Hem” was used as positive control. The arrows show activity-stained bands of trypsin inhibitors and chymotrypsin inhibitors, which are named “TI-1~TI-6” and “CI-1~CI-6” from top to bottom.

**Figure 4 molecules-27-01820-f004:**
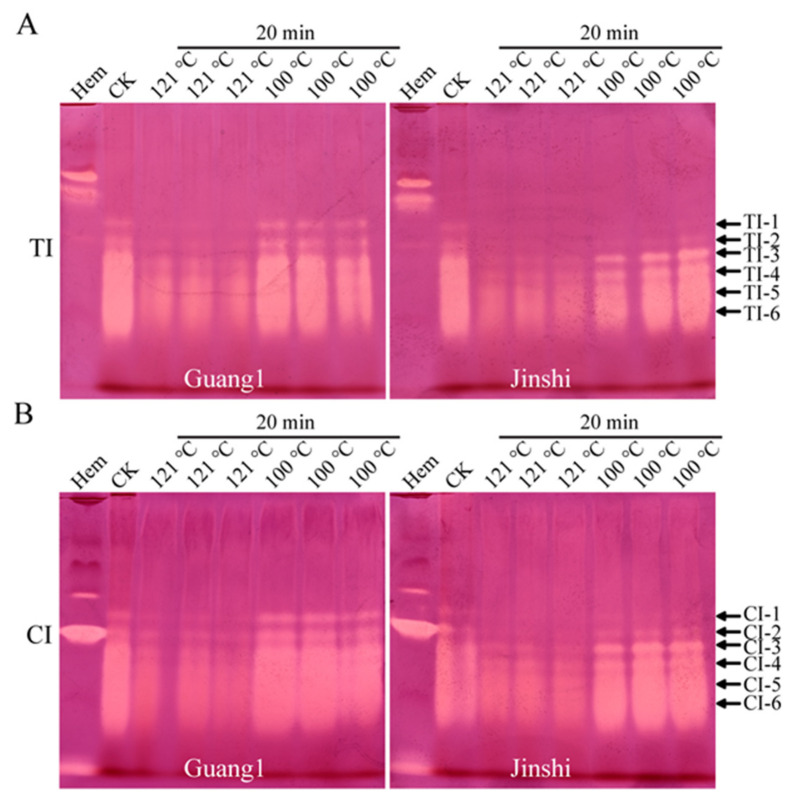
High temperature combined with high pressure could greatly weaken the activities of TIs and CIs from mulberry leaves. Activity staining of TIs (**A**) and CIs (**B**) from leaves of mulberry varieties Guang1 and Jinshi based on alkaline Native PAGE. “TI” and “CI” represent trypsin inhibitor and chymotrypsin inhibitor, respectively. B. mori hemolymph from day-5 fifth-instar larvae represented by “Hem” was used as positive control. “121 °C” and “100 °C” show the mulberry-leaf proteins were incubated at 121 °C or 100 °C for 20 min. The arrows show activity-stained bands of trypsin inhibitors and chymotrypsin inhibitors, which are named “TI-1~TI-6” and “CI-1~CI-6” from top to bottom.

**Figure 5 molecules-27-01820-f005:**
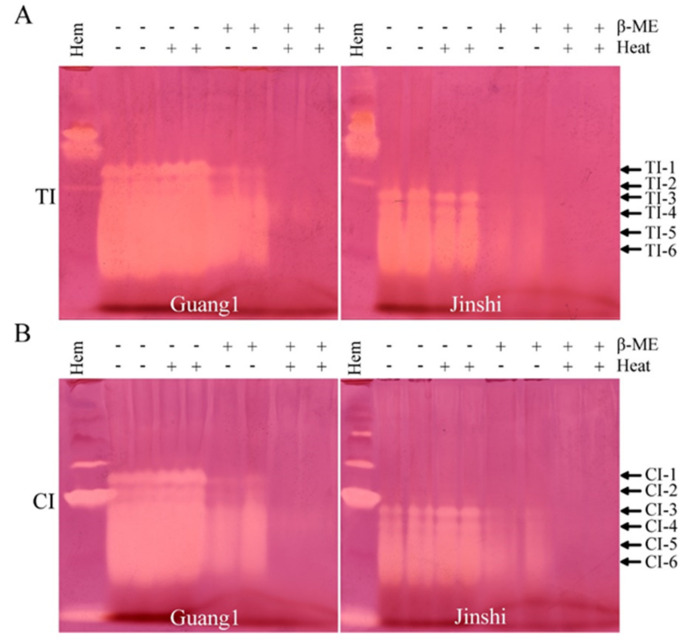
β-mercaptoethanol treatment could completely eliminate the activities of TIs and CIs from leaves of Guang1 and Jinshi. Activity staining of TIs (**A**) and CIs (**B**) from leaves of mulberry varieties Guang1 and Jinshi based on alkaline Native PAGE. “TI” and “CI” represent trypsin inhibitor and chymotrypsin inhibitor, respectively. *B. mori* hemolymph from day-5 fifth-instar larvae represented by “Hem” was used as positive control. “β-ME” stands for reducing agent β-mercaptoethanol. “+” or “-” indicates with or without β-mercaptoethanol and heat treatment. The arrows show activity-stained bands of trypsin inhibitors and chymotrypsin inhibitors, which are named “TI-1~TI-6” and “CI-1~CI-6” from top to bottom.

**Figure 6 molecules-27-01820-f006:**
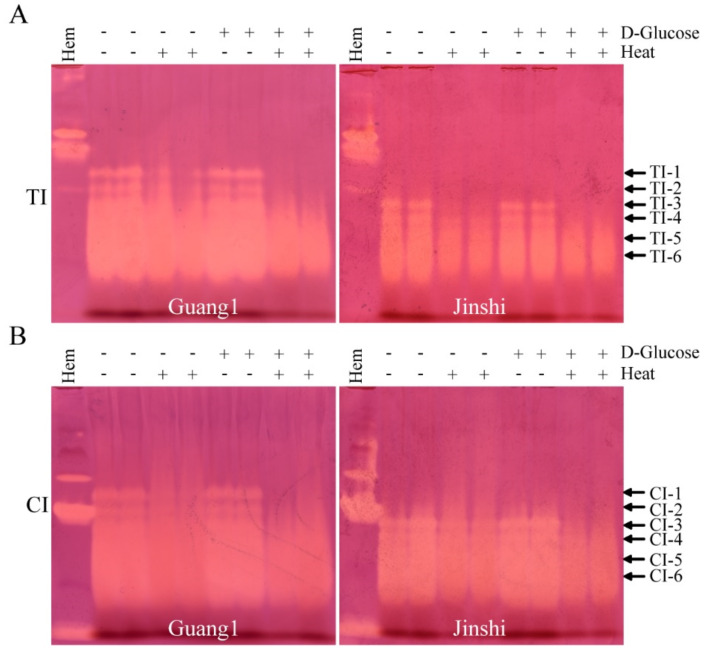
Maillard reaction could weaken the activities of TIs and CIs from leaves of Guang1 and Jinshi. Activity staining of TIs (**A**) and CIs (**B**) from leaves of mulberry varieties Guang1 and Jinshi based on alkaline Native PAGE. “TI” and “CI” represent trypsin inhibitor and chymotrypsin inhibitor, respectively. *B. mori* hemolymph from day-5 fifth-instar larvae represented by “Hem” was used as positive control. “+” or “-” indicates with or without D-Glucose and heat treatment. The arrows show activity-stained bands of trypsin inhibitors and chymotrypsin inhibitors, which are named “TI-1~TI-6” and “CI-1~CI-6” from top to bottom.

**Table 1 molecules-27-01820-t001:** Mulberry varieties used in the experiments.

No.	Cultivars	Species	No.	Cultivars	Species
1	Nong14	*Morus multicaulis*	18	Dongguangdabai	*Morus alba*
2	Nong12	*M. multicaulis*	19	Zhenzhubai	*M. alba*
3	Sichuantiansang	*M. multicaulis*	20	Guoshensang	*M. alba*
4	Luyou3	*M. multicaulis*	21	Xiaoguansang	*M. alba*
5	Luyou7	*M. multicaulis*	22	Shaansang403	*M. alba*
6	Heiyumodou	*M. multicaulis*	23	Qingyeshufan	*M. alba*
7	Yu711	*M. multicaulis*	24	Xiaohongpi	*M. alba*
8	Dashi	*Morus atropurpurea*	25	Huaiyin4	*M. alba*
9	Hongguo2	*M. atropurpurea*	26	Shaosang	*M. alba*
10	Lunjiao408	*M. atropurpurea*	27	Tuosang	*M. alba*
11	Beiqu1	*M. atropurpurea*	28	Ukrainian1	*M. alba*
12	Shalun	*M. atropurpurea*	29	Xinjiangbaisang	*M. alba*
13	Guang1	*M. atropurpurea*	30	Jinshi	*Morus mizuho*
14	Guangdongzajiao	*M. atropurpurea*	31	Hulusang	*Morus cathayana*
15	Guiyou12	*M. atropurpurea*	32	Yixianheisang	*Morus nigra*
16	Guiyou62	*M. atropurpurea*	33	Taiwanchangguosang	*Morus leavigata*
17	Shi2	*M. atropurpurea*	34	Kenmochi	*Morus bombycis*

## Data Availability

Not applicable.

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
