# Peer review of "Physicochemical Properties and Elimination of the Activity of Anti-Nutritional Serine Protease Inhibitors from Mulberry Leaves"

_molecules, 2022, doi:10.3390/molecules27061820_

Round 1

Reviewer 1 Report

Authors should state clearly the physical state of the leaf samples used for extraction, i.e. whether fresh leaves were used for extraction or dried leaves?

Author Response

Comments and Suggestions for Authors

  1. Authors should state clearly the physical state of the leaf samples used for extraction, i.e. whether fresh leaves were used for extraction or dried leaves?

Response:

Thanks for your valuable suggestions. Fresh mulberry leaves were washed, drained and then ground into powder in liquid nitrogen for protein extraction. The details have been added to material and methods (Page 7 line 323).

  1. In the revised manuscript,we have done our best to improve the English writing of paper.

Reviewer 2 Report

Luo et al wish to report their finding of the presence of serine protease inhibitors in mulberry leaves. The authors have systematically examined the leaves from 34 mulberry varieties and show that these leaves have different levels of trypsin inhibitors (TI-1 to TI-6) and chymotrypsin inhibitors (CI-1 to CI-6). They have also studied the physicochemical properties of these inhibitors and show that these inhibitors have strong acid and alkali resistance and high thermal stability. The authors have examined different conditions to reduce the activity of these protease inhibitors and demonstrate that beta-mercaptoethanol treatment is effective to inhibit protease inhibitors activity and potentially improve the nutritional value of mulberry leaves. Overall, this is an interesting study and merits publication in Molecules after addressing the below comments.

  1. The authors mentioned that they identified inhibitors TI-1 to TI-6 and CI-1 to TI-6 in mulberry leaves. However, key information regarding their identity is missing. Can the authors provide information like the name/molecular weight of these inhibitors?  
  2. The authors observed different levels of trypsin and chymotrypsin inhibitors in different varieties of mulberry leaves. Also, different levels were observed across the different cultivars in the same species. Can the authors comment on possible reasons why these inhibitors levels are different? 
  3. The authors indicated six trypsin inhibitors (TI-1 to TI-6) and six chymotrypsin inhibitors (CI-1 to CI-6) in mulberry leaves. However, in gel images- there are smears for some variety of mulberry leaves. For example, Gaung1, Shaosang, Jinshi. If these bands are resolved, how authors can claim only 6 TIS and 6 CIs? 
  4. Figure 2. The line below the species name is not shown correctly. For example, Dashi’s species name is Morus atropurpurea. However, it is shown under the M. multicaulis.  

Author Response

Thanks for your careful reading of the manuscript and helpful comments and suggestions. In revising the paper, we have carefully considered your comments and suggestions. We are trying our best to answer your questions.

Comments and Suggestions for Authors

  1. The authors mentioned that they identified inhibitors TI-1 to TI-6 and CI-1 to CI-6 in mulberry leaves. However, key information regarding their identity is missing. Can the authors provide information like the name/molecular weight of these inhibitors?  

Response:

Thanks for your kind suggestion. In this study, we confirmed the presence of multiple TIs and CIs activities in mulberry leaves, and named them TI-1 to TI-6 and CI-1 to CI-6 from top to bottom based on the mobility of these protease inhibitors in alkaline gel. Information like the name/molecular weight of these inhibitors is not yet available. In future studies, we plan to cut the gel according to the position of TI-1 to TI-6 and CI-1 to CI-6 in the gel, and perform mass spectrometry identification and cloning of protease inhibitors in the gel slices. The physicochemical properties of these inhibitors and their physiological functions in mulberry leaves were deeply investigated.

  1. The authors observed different levels of trypsin and chymotrypsin inhibitors in different varieties of mulberry leaves. Also, different levels were observed across the different cultivars in the same species. Can the authors comment on possible reasons why these inhibitors levels are different?

Response:

Thank you very much for your professional review and your kindly suggestion. 34 mulberry varieties, including 30 cultivated species and 4 wild species, are cultivated in Hengkou Sericulture Research Base, Ankang University, China. The mulberry leaves were picked on August 19, 2020. Maturity of the mulberry leaves was basically the same. There were differences in the activity levels of protease inhibitors across different mulberry cultivars in the same species, which might be due to the large genetic differences among different mulberry cultivars. The research shows that mulberry germplasm resources are extremely rich and complex, which also results in different traits among different mulberry, such as disease resistance, insect resistance, stress resistance, etc(Xuan et al., 2022).

  1. The authors indicated six trypsin inhibitors (TI-1 to TI-6) and six chymotrypsin inhibitors (CI-1 to CI-6) in mulberry leaves. However, in gel images- there are smears for some variety of mulberry leaves. For example, Gaung1, Shaosang, Jinshi. If these bands are resolved, how authors can claim only 6 TIs and 6 CIs?

Response:

The activities of TI-1 to TI-2 and CI-1 to CI-2 were weak, while those of TI-3 to TI-6 and CI-3 to CI-6 were very strong. The mobility of TI-3 to TI-6 and CI-3 to CI-6 in Native PAGE is relatively close. It is difficult to completely separate the active bands of TI-3 to TI-6 and CI-3 to CI-6, while ensuring the detection of TI-1 to TI-2 and CI-1 to CI-2. TIs and CIs in mulberry leaves can be named by comprehensively comparing the active band positions of TIs and CIs in various varieties.

  1. Figure 2. The line below the species name is not shown correctly. For example, Dashi’s species name is Morus atropurpurea. However, it is shown under the multicaulis.

Response:

We have modified Figure 2 in the revised manuscript (Page 11 line 482).

References

Xuan, Y.; Ma, B.; Li, D.; Tian, Y.; Zeng, Q.; He, N. Chromosome restructuring and number change during the evolution of Morus notabilis and Morus alba. Horticulture Research. 2022, 9, uhab030.